# The value of long-duration energy storage under various grid conditions in a zero-emissions future

Martin Staadecker [1,2] ✉, Julia Szinai [3], Pedro A. Sánchez-Pérez[4], Sarah Kurtz[5] & Patricia Hidalgo-Gonzalez [1,6] ✉

Long-duration energy storage (LDES) is a key resource in enabling zero-emissions electricity grids but its role within different types of grids is not well understood. Using the Switch capacity expansion model, we model a zero-emissions Western Interconnect with high geographical resolution to understand the value of LDES under 39 scenarios with different generation mixes, transmission expansion, storage costs, and storage mandates. We find that a) LDES is particularly valuable in majority wind-powered regions and regions with diminishing hydropower generation, b) seasonal operation of storage becomes cost-effective if storage capital costs fall below US$5 kWh$^{-1}$, and c) mandating the installation of enough LDES to enable year-long storage cycles would reduce electricity prices during times of high demand by over 70%. Given the asset and resource diversity of the Western Interconnect, our results can provide grid planners in many regions with guidance on how LDES impacts and is impacted by energy storage mandates, investments in LDES research and development, and generation mix and transmission expansion decisions.

The International Panel on Climate Change recommends limiting net emissions to zero by 2050[1] and 77 countries have set such a goal[2]. Considering that the electricity sector is responsible for roughly a third of global greenhouse gas (GHG) emissions[3], policy makers are particularly interested in the question of how to decarbonize electricity grids. The rapid rise in renewable energy driven by declining technology costs[4] and incentive programs[5–7] has proven to be an effective way to reduce GHG emissions in the electricity sector. However, reaching an entirely net zero and reliable electricity grid remains challenging[8–10]. Numerous potential solutions to this problem have been studied by researchers[11]. Some studies focus on "firm" low-carbon resources capable of flexible operation[12–16]. Other studies focus on transmission expansion and inter-regional coordination as a source of flexibility[17,18]. Finally, given the consistent cost declines in storage technologies[19] and the expectation that they will continue[20], several

studies explore the role of short-duration energy storage and long-duration energy storage (LDES) in providing flexibility to the grid[21–30]. In this paper, we follow the emerging trend[31,32] of defining LDES as any type of storage with 10 or more hours of duration. Conversely, short-duration storage is defined as any type of storage with fewer than 10 h of duration. We also define seasonal storage—a subset of LDES—as any type of storage that is operated such that charge-discharge cycles occur over several months.

Prior studies of LDES primarily evaluate the role that different LDES technologies could play in decarbonizing the power system by performing LDES cost sensitivities or LDES parameter sensitivities (e.g., charging/discharging efficiencies), and identifying the impact that LDES deployment has on low- or zero-emission electricity grids. However, existing studies are limited because of modeling simplifications such as the exclusion of longer duration storage and the

[1]Mechanical and Aerospace Engineering, University of California San Diego, La Jolla, CA, USA. [2]Division of Engineering Science, University of Toronto, Toronto, ON, Canada. [3]Lawrence Berkeley National Laboratory, Berkeley, CA, USA. [4]National Renewable Energy Laboratory, Golden, CO, USA. [5]School of Engineering, University of California Merced, Merced, CA, USA. [6]Center for Energy Research, University of California San Diego, La Jolla, CA, USA. ✉e-mail: staadeck@mit.edu; phidalgogonzalez@ucsd.edu

exclusion of multi-nodal transmission networks. For example, Sisternes et al.[23] studies the value of up to 30 gigawatts (GW) of 2- or 10-h storage in providing up to 90% $CO_2$ emission reductions (from 2013 levels) to the Texan electricity grid (ERCOT) in 2035. Although they simultaneously optimize capacity expansion and hourly unit commitment, they do not model >10 h storage resources, the transmission network, and a diversity of wind and solar resources. To understand the value of >10 h storage, Dowling et al.[24] study a 100% renewable energy grid using only solar, wind, li-ion short-duration storage, and LDES. They find that LDES duration increases from ~400 to ~700 h as more years of weather data, i.e., weather years, are considered. However, as the authors mention, their findings correspond to a lower limit of potential benefits of LDES deployment because of modeling simplifications. Simplifications include averaging solar and wind capacity factors across the USA, not considering the effect of electrification on electricity demand growth, and not modelling transmission (singe region), its losses or its expansion. The work in Guerra et al.[26] and a follow up study[27] are some of the only LDES studies that model transmission lines. The first study models the Western US grid using an aggregated representation of transmission lines with up to 83% of variable renewable energy. However, the modeling approach they use consists of running a capacity expansion model with a posterior simulation using a production cost model where LDES is omitted. LDES is only included at a later phase when the authors use a revenue model. This framework does not allow for the understanding of the optimal deployment and operation of LDES as a function of several relevant factors of the grid (e.g., hydropower availability, restricting transmission expansion, etc.). The follow up study by Guerra et al.[27] analyzes seven independent system operators in the USA to determine the level of renewable penetration at which LDES is deployed in a cost minimization. The aim of Sepulveda et al.[25] is to systematically understand the design space for different parameters associated with LDES (e.g., cost, charging, and discharging efficiencies) using a capacity expansion model for New England and Texas and different combinations of LDES design parameters. A summary of the comparison of these papers and the current paper is shown in Table 1.

In summary, existing work that studies LDES in the context of decarbonization of the US grid focuses on understanding the variation in LDES as more weather years are considered, the value of LDES to the grid as a function of decarbonization levels, the technologies LDES can displace, the cost targets LDES must reach to become competitive, and the key technical parameters of LDES that are the most critical from a power systems operations perspective. However, to date, there is still a need for studies that analyze the other side of the equation: how the composition of the generation mix and the physical and policy characteristics of the grid (e.g., solar or wind dominant, transmission constraints, storage mandates, etc.) affect the deployment and operation of LDES.

To address these research gaps, this study conducts a systematic analysis that identifies and studies four of the most important characteristics of the grid that affect the value and optimal deployment and operations of technology-agnostic LDES in a zero-emissions power system. First, we study how the deployment and operation of LDES is impacted by the share of solar and wind generation capacity (i.e., solar-dominant versus wind-dominant grids). Solar and wind dominant grids are expected to require different storage durations since solar has a diurnal cycle and wind might not. Studying how the use and value of LDES changes under a range of solar to wind-dominant grids enables policymakers to make geographically appropriate decisions that are robust to a range of wind and solar cost futures. Second, we study the impact of decreases in hydropower generation on LDES. Hydropower generation patterns are changing under the effects of climate change[33–35] which puts grid operators at risk of losing part of their hydropower capacity—one of the largest zero-emissions sources of flexibility in the US grid[36]. Hence, if hydropower availability decreases, this will impact the need of other balancing sources such as LDES. Third, we study the impact on LDES of transmission deployment—another key source of flexibility for balancing electricity grids[17,37]. Hence, understanding the impacts on LDES deployment as we explore upper and lower bounds for transmission deployment can provide meaningful insights for policy makers considering the political and environmental hurdles to build transmission lines. Fourth, there is value and a lack of understanding around the impact and potential benefits of LDES energy capacity mandates. In this context, we refer to an LDES energy capacity mandate as a quantity of storage energy capacity that is mandated by a governmental entity to be built by 2050 across Western North America. Power capacity storage mandates have had an important role; for example, California was the first state to have power capacity storage mandates to support grid decarbonization[38]. This initiative has had multiplicative effects in solidifying the market for storage investors and suppliers, encouraging more research and development, and producing operational benefits for the grid as more renewables are integrated. Several states have since followed this initiative[39]. In this study, we focus on evaluating the design of possible future storage energy capacity mandates instead of power capacity mandates because we want to understand the energy balancing benefits of such mandates and their impacts for the grid (e.g., electricity pricing impacts, curtailment and operational impacts, zonal distribution of optimal LDES placement, etc.). This is a research question that is important for policy makers to explore as a pathway to support the decarbonization of our grids.

The Western Electricity Coordinating Council (WECC) region is an appropriate case study to address these questions as it is the largest US interconnection with very diverse and rich renewable sources, which we model with more than 8000 geolocated sites for candidate wind and solar deployment. To the best of our knowledge, this model has one of the higher geographical resolutions to represent renewable sources among current US capacity expansion models. Modeling the Western Interconnect as a transmission network with 50 zones also enables us to study the contribution of geographic variation of the above grid characteristics to the deployment and operations of LDES.

We model the Western Interconnect with a 2050 zero-emissions future using Switch[40], a long-term capacity expansion model that has been used in numerous studies of low- or zero-emissions electricity grids[30,41–43]. Our model contains 3580 existing plants of which 1010 are expected to still be in service by 2050 (hydropower and nuclear). Our model also contains 4908 candidate plants that the model may choose to build (on-shore wind, off-shore wind, various types of solar, biomass, and geothermal). These plants are distributed across 50 load zones that cover the WECC and are connected by 126 aggregated transmission lines (Supplementary Fig. 1). The model simultaneously optimizes investment and dispatch decisions to minimize the total system cost and meet each load zone's power demand while considering the transmission network. Dispatch decisions are made at consecutive 4-h intervals throughout every day of the 2050 year. No temporal aggregation techniques were used (e.g., "representative weeks"[23]). The use of a 4-h interval instead of the typical hourly dispatch is part of the reason high geographic resolution could be achieved. We quantify in the "Methods" section the impact of the lower 4-h temporal resolution on results by comparing it to a sensitivity run with a 1-h temporal resolution.

To model storage, each load zone contains one candidate storage plant with fixed cost and performance parameters (Table 2). We complete sensitivity runs of alternate cost scenarios (Table 3); however, the focus of the study is the interaction of LDES with the electricity grid since an in-depth study of LDES cost and performance parameters has already been performed[25]. Energy and power capacity of candidate storage plants are unconstrained and optimized by the model from the perspective of the grid, such that the model may build storage of any duration and size in each load zone. The model also

**Table 1 | Comparison of papers studying storage with cost-minimization models**

| | Sisternes et al.[23] | Guerra et al.[26] | Dowling et al.[24] | Sepulveda et al.[25] | This paper |
|---|---|---|---|---|---|
| Highlights of study | Storage modelling with unit commitment and capacity expansion | Seasonal storage with transmission constraints modelling | Impact of LDES in a solar, wind and batteries grid, with multi-year optimization | Impact of LDES with thousands of cost and performance scenarios and firm generation | Impact of various generation mixes, transmission deployments, and energy capacity storage mandates on LDES |
| Time period modelled | 2035 | 2024–2050 | 1980–2018 | 2045 | 2050 |
| Storage technologies modelled | Li-ion, PHS | H₂, PHS, CAES | Li-ion, H₂, PHS, CAES | Technology-agnostic LDES | Technology-agnostic LDES, existing batteries and PHS[a] |
| Model optimized storage duration | ✗ 2 h or 10 h only | ✗ 1 d, 2 d, 1 w, 2 w, 1 m only | ✓ | ✓ | ✓ |
| Model optimized storage power capacity | ✗ Exogenously added ≤ 30 GW of storage | ✗ Exogenously added 2 GW of storage | ✓ | ✓ | ✓ |
| Modelled firm low- or no-carbon technologies | ✓ Nuclear | ✓ Nuclear, biomass, hydro, geothermal | ✗ | ✓ Nuclear, natural gas + CCS, blue H₂ | ✓ Pre-existing nuclear, biomass, geothermal, hydro |
| Modelled unit commitment | ✓ | ✓ | ✗ | ✓ | ✗ |
| Transmission lines modelled | 0 | 77 | 0 | 0 | 126 |
| Balancing regions modelled | 1 ERCOT | 35 WECC | 1 Contiguous US[b] | 1 Texas or New England[c] | 50 WECC |
| Co-optimized capacity expansion and dispatch | ✓ | ✗ LDES modelled after capacity expansion | ✓ | ✓ | ✓ |
| Modelled existing generation capacity | ✗ Greenfield | ✓ | ✗ Greenfield | ✗ Greenfield | ✓ |
| Accounted for changing demand patterns due to electrification | ✗ Scaled demand by 1.86% annually | ✗ Scaled demand using regional growth factors | ✗ Used historical loads to model 1980–2018 | ✓ Scaled demand by 1% annually. High-electrification scenarios used an adjusted load profile | ✓ Scaled demand accounting for increased energy efficiency, building electrification and zero emission vehicles |
| Zero-emissions study | ~60–90% emission reductions | 80% renewable portfolio standard | ✓ | ✓ | ✓ |
| Hourly temporal resolution | ✓ | ✓ | ✓ | ✓ | ~[d] |
| ≥ 1 year storage balancing horizon | ✗ 1 week | ✓ 1 year | ✓ Up to 6 years | ✓ 1 year | ✓ 1 year |

*PHS* pumped storage hydropower, *CAES* compressed air energy storage, *LDES* long-duration energy storage.
[a]See limitation section for limitations on pre-existing PHS modelling.
[b]Sensitivity analyses considered additional regions (ERCOT, WECC, and the Eastern Interconnect).
[c]The authors mention that New England and Texas are not modelled "with realism." Rather, these regions represent a Northern-like and Southern-like grid.
[d]4 h resolution for all runs except for one sensitivity run with hourly resolution.

optimizes, from the grid's perspective, charging and discharging decisions throughout the year 2050 at 4-h intervals which allows storage plants to operate with cycles ranging from 8 hours up to 1 year.

## Table 2 | Candidate storage baseline parameters

| Storage parameter | Value |
|---|---|
| Power capacity (MW) | Unconstrained variable optimized by the model to minimize system cost |
| Energy capacity (MWh) | Unconstrained variable optimized by the model to minimize system cost |
| Overnight power installation costs | 19.58 $/kW |
| Overnight energy installation costs | 22.43 $/kWh |
| Yearly fixed O&M costs | 6.096 $/kW-yr |
| Variable O&M costs | None |
| Round-trip efficiency | 85% |
| | (92% charging efficiency, 92% discharging efficiency) |
| Charging rate (MW) | Equal to discharging rate (power capacity) |
| Lifetime | 15 years |
| Idle Losses (self-discharge rate) | None |
| Scheduled outage rate | 0.0055 |
| Forced outage rate | 0.02 |

We run our model under a baseline scenario and 38 alternate scenarios, where we vary five main attributes to understand the value and role of LDES: wind-vs-solar capacity shares, hydropower availability, transmission expansion costs, storage energy capacity costs, and storage mandates (labeled A to E as defined in Table 3). The scenarios are chosen to explore parameters that would impact the use of storage, or parameters that could change unexpectedly as the WECC transitions to a zero-emissions grid (see Table 3 for further justification of the scenario selection).

In this work, we perform a systematic study of the different grid factors that affect the value of LDES in a zero-emissions Western North American grid. We find that 6-to-10-h duration storage assets optimally support Southwest regions that are solar-dominant. On the other hand, wind-dominant regions are better supported by 10-to-20-h storage assets. In hydropower-dominant regions, a 50% reduction in hydropower availability would cause the average storage duration to shift from 6.3 to 23 h. Disallowing transmission expansion results in 32% more storage energy capacity being required compared to the baseline. Depending on the overnight cost assumed for storage energy capacity we observe a range of optimal maximum duration starting from 9 to ~800 h (where transmission deployment decreases by 75%). From analyzing 13 possible LDES mandates for the WECC, we identify that the mandate that brings the highest relative value to the grid corresponds to 20 terawatt hours (TWh) of installed storage energy capacity by 2050. We find that this mandate would reduce curtailment by 92%, total installed power capacity by 10%, transmission deployment by 75%, and electricity prices during peak periods by 70%.

## Table 3 | The baseline scenario and 38 alternate scenarios grouped in 5 sets

| Scenario set | Description of zero-emissions scenario |
|---|---|
| Baseline model of the Western Electricity Coordinating Council (WECC) | The baseline scenario models a 2050 zero-emissions future. It uses NREL's 2020 Annual Technology Baseline (ATB)[56] cost projections for 2050 and a high-electrification high-energy-efficiency demand scenario. Storage power and energy capacity costs are 19.58 $/kW and 22.43 $/kWh, respectively, with O&M costs of 6.10 $/kW-year. These costs represent a scenario where NREL's lithium-ion 2020 storage costs drop by 90% by 2030 per the U.S. Department of Energy's (DOE) Energy Storage Grand Challenge, and then drop further between 2030 and 2050 at the same rate as the moderate projections in NREL's 2020 ATB. |
| Set A: Varying wind-vs-solar capacity shares | Set A compares the baseline to 8 scenarios where an extra constraint fixes the ratio of total wind capacity to total solar capacity in the WECC. The ratio constraint has a capacity expansion perspective, it does not force early retirements to achieve the desired ratio. This enables the study of a WECC with wind-vs-solar shares varying from 91% solar and 9% wind to 40% solar and 60% wind. For reference, the baseline scenario's wind-vs-solar share is 81% solar and 19% wind. This set is of interest since wind and solar are projected to be dominant technologies in a zero-emissions WECC[27] but their expected relative share of the generation mix might change as technology costs and system characteristics evolve. |
| Set B: Reduced hydropower generation | Set B compares the baseline to 5 scenarios where hydropower generation is limited in the WECC by derating monthly average water flows at all hydropower plants equally. Water flows are derated by anywhere from 15% to 100% depending on the scenario. This set is of interest since hydropower generation patterns are changing under the effects of climate change[33–35] and we wish to understand how these changes, and more extreme ones, might impact long-duration energy storage (LDES). This set with decreases in hydropower generation can also be considered more generally representative of a future where additional LDES may need to be built to compensate for a loss of flexible, zero-emissions resources. |
| Set C: Varying transmission expansion costs | Set C compares the baseline to two different transmission expansion scenarios. The first scenario represents a grid with limited transmission expansion (by increasing the cost of expanding transmission lines tenfold). The second scenario represents a grid without transmission congestion due to unlimited transmission capacity (by setting the cost of expanding transmission lines to zero). This second scenario is not a "copperplate" scenario since transmission losses still occur. Both scenarios represent opposite extremes and therefore provide bounds on the behavior of future grids. These bounds are particularly useful since transmission expansion is difficult to model due to its dependence on political and social factors that are not captured in a purely cost-based model. |
| Set D: Varying storage energy capacity costs | Set D compares the baseline where storage energy capacity costs are 22.43 $/kWh to 10 scenarios where storage energy capacity costs range from 0.5 to 102 $/kWh. The upper bound of 102 $/kWh corresponds to NREL's 2021 ATB moderate scenario cost projections for utility-scale battery in 2050[4]. This set is of interest since storage energy capacity costs are one of the greatest determinants of LDES deployment[25] and may vary significantly depending on the development of various LDES technologies. |
| Set E: WECC under different LDES mandates | Set E compares the baseline that has a total of 1.94 TWh of storage energy capacity in the WECC to 13 scenarios where an extra constraint increases the total WECC storage energy capacity to anywhere from 2 to 64 TWh. This constraint represents a WECC-wide energy storage mandate. Using this region-agnostic approach to study storage mandates, policy makers can evaluate the impact of mandates on electricity pricing and grid behavior. |

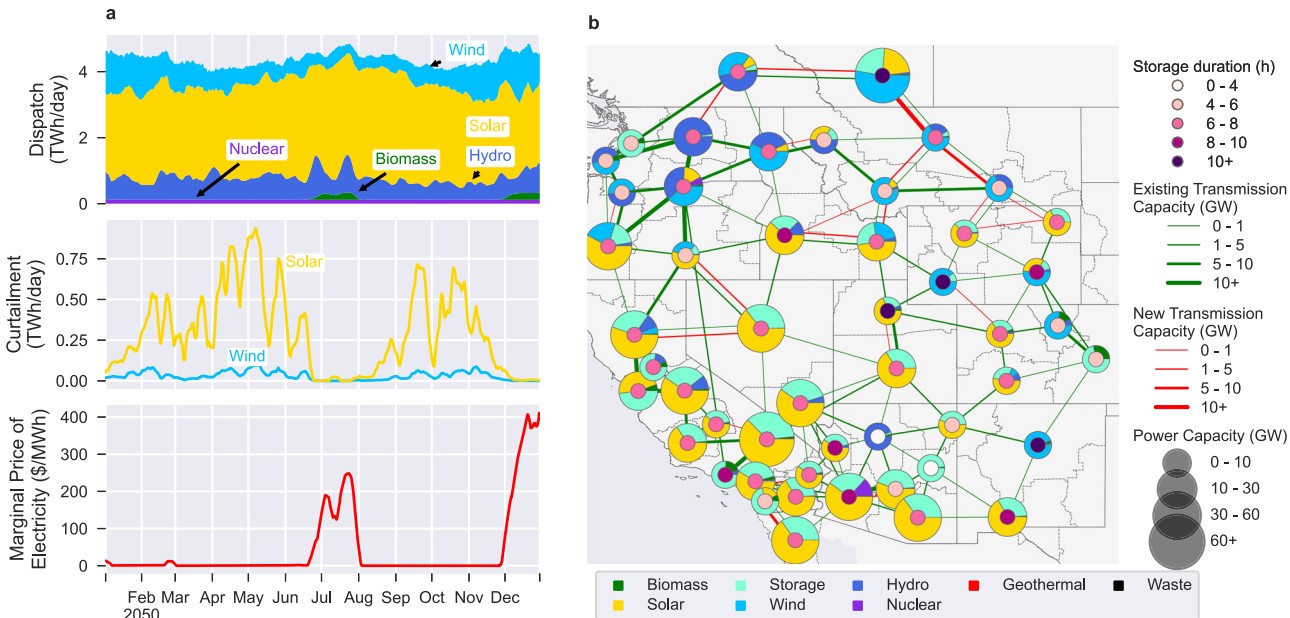

**Fig. 1 | The Western Interconnect in a baseline 2050 zero-emissions future.** The baseline scenario represented in time (**a**) and space (**b**) highlighting seasonal variations in the dispatch by technology, curtailment and mean marginal electricity price (**a**) as well as differences between the southern and northern Western Interconnect (**b**). Temporal values in a are 7-day rolling averages, storage durations are per-load-zone averages, and power capacity is the total combined sum of the installed generation and storage capacity.

## Results and discussion

### The 2050 zero-emissions baseline Western Interconnect

In this section, we analyze the baseline scenario which represents a least-cost zero-emissions WECC. We find seasonal and geographical trends in generation and storage technology use. We also find that the baseline scenario uses little LDES. Excluding Alberta, which holds 300 GW of 18-h storage, the baseline's energy storage is 99% short-duration energy storage (under 10 h duration).

Throughout this paper, we reference the marginal price of electricity. Marginal prices are calculated from the dual values of the energy balance constraint for each time point in each load zone in our linear program. Since this constraint specifies that the total generation must equal the total load in each load zone at a certain time, the constraint's dual values are the marginal prices of supplying additional power at that time. Marginal prices in this work are similar but not equivalent to the typically quoted market locational marginal prices since our marginal price values represent not only marginal operating costs but also marginal investment costs.

For most of the year, solar curtailment (Fig. 1) is high (up to 33% of solar generation curtailed in a week). However, demand increases during the mid-summer months and early winter months causing solar and wind curtailment to drop to zero. During these peak months, there is a sharp increase in the marginal electricity price with the 7-day average marginal price reaching 250 US-dollars per megawatt hour ($/MWh) in late July and 410 $/MWh in late December. Finally, wind generation is highest and solar generation is lowest during the winter. Specifically, the average daily generation before curtailment from November to March (inclusive) compared to the rest of the year is 98% higher for wind and 16% lower for solar.

Additionally, under the baseline scenario, the southern WECC (Baja Mexico and states: California, Nevada, Utah, Colorado, Arizona, New Mexico) and the northern WECC (Canada and states: Washington, Oregon, Idaho, Montana, Wyoming) rely on different generation and storage technologies. As shown in Fig. 1, in the southern WECC, solar power is the dominant technology and is used to recharge 6-to-8-h duration energy storage that provides power when the sun is not shining. In the northern WECC, the grid relies primarily on a mix of hydro and wind power coupled with greater transmission capacity. Storage duration varies more significantly in the northern WECC compared to the nearly unvarying 6-to-8-h duration of the southern WECC. In the northern WECC, storage duration varies from less than 6 h in multiple load zones to 18 h in the Alberta load zone.

This least-cost investment and operational plan for a decarbonized WECC in 2050 relies on regional coordination highlighting the role and importance that the Western Energy Imbalance Market could have. On the other hand, strong regional coordination, thus dependency, may exacerbate reliability challenges during extreme weather events as neighboring regions might not be able to provide power as expected during normal conditions.

### Factors impacting the value of LDES

The value of LDES is closely tied to the composition and characteristics of the rest of the energy grid. In this section, we share results on how four key factors (wind-vs-solar capacity shares, hydropower availability, transmission expansion and energy storage costs) impact the value of LDES.

When varying the relative proportion of wind-vs-solar capacity (scenario set A), we find that LDES is more valuable in wind-dominant grids than in solar-dominant grids. We find that the power capacity of 6-to-10-h storage in the scenarios is roughly proportional to the solar capacity (Fig. 2). In the most solar-dominant scenario (91% solar, 9% wind, i.e., five times more solar than wind), the WECC has 243 GW of 6-to-10-h storage and this amount drops roughly linearly to 97 GW In the most wind-dominant scenario (40% solar, 60% wind) (Supplementary Fig. 2). This relationship suggests that 6-to-10-h storage is the ideal duration to support the diurnal cycles of solar power. In wind-dominant scenarios, 6-to-10-h storage is replaced by 10-to-20-h storage that appears better suited to support wind-dominant grids. A closer look at the distribution of storage resources in a solar-dominant and wind-dominant scenario (Fig. 3) confirms that nearly all solar-dominant load zones use 6-to-10-h storage, while nearly all wind-dominant load zones use 10-to-20-h storage.

We also find that the lowest levels of transmission expansion (16 million new MW-km) among the scenarios in set A occur in the

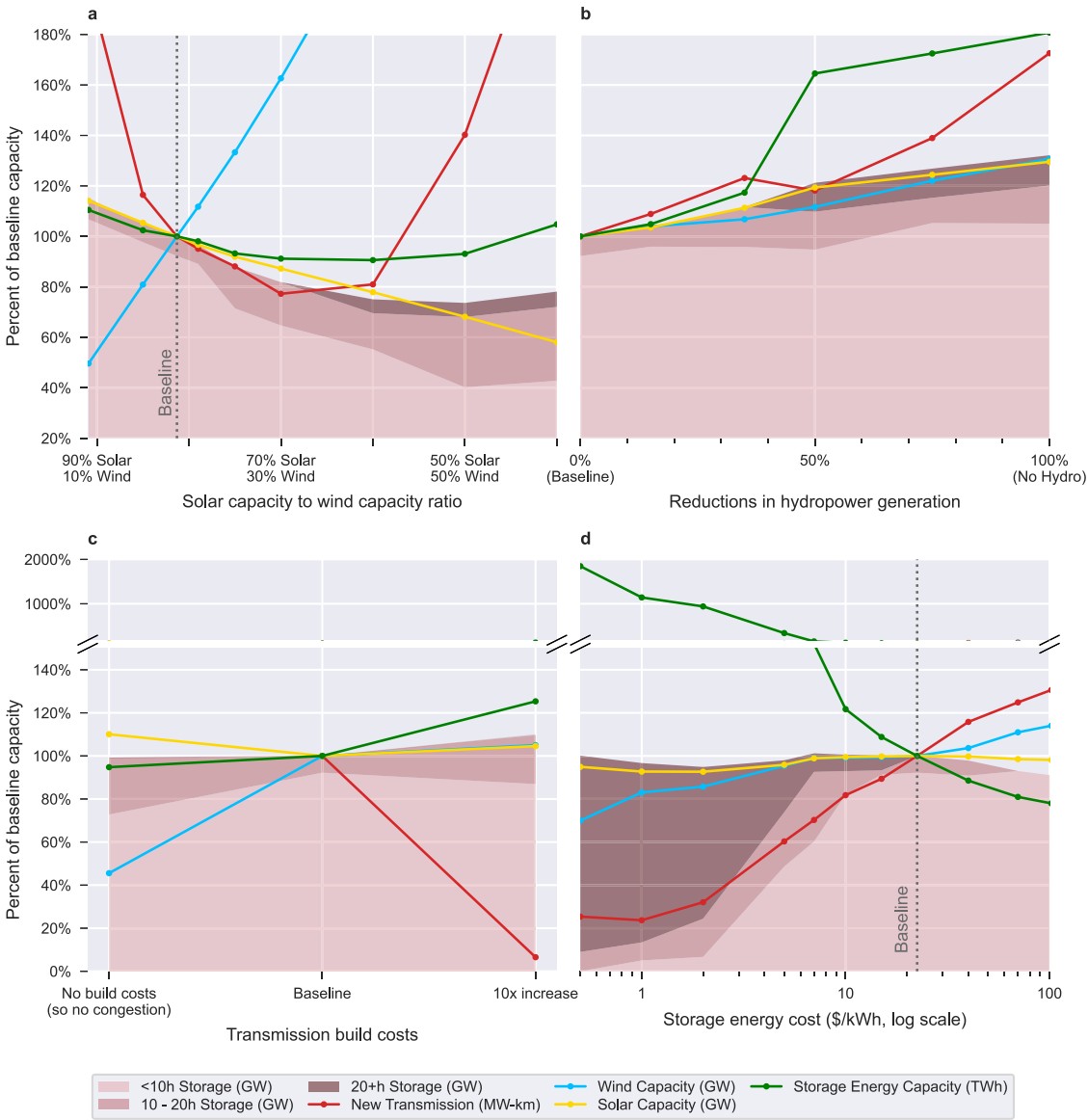

**Fig. 2 | Impact of 4 key factors on the use and value of LDES relative to the baseline.** The impact relative to the baseline of variations in four key parameters (**a**–**d**) on the storage power capacity (area plot), storage energy capacity (green line, TWh), wind capacity (blue line), solar capacity (yellow line), and transmission expansion (red line). The transmission expansion line in (**c**) does not extend to the "No Tx Build Costs" scenario since that scenario has unlimited transmission capacity.

70%-solar 30%-wind scenario (Fig. 2a), implying that the solar and wind resources of this scenario are best able to serve their local load zones. This result hints at the natural availability of solar and wind resources in the WECC. However, this scenario is not the least-cost scenario since it is cheaper to expand transmission and import power from regions with lower power costs.

We analyze the impact of reductions in hydropower dispatch (scenario set B) and find that any reduction beyond the historical averages used in the baseline causes a need for storage of longer durations and greater capacities (Fig. 2b). A 50% reduction in hydropower generation increases the WECC-wide storage energy and power capacity by 65% and 21%, respectively. Further, this reduction shifts the average storage duration from 6.3 to 23 h in the six load zones where hydropower was previously responsible for most of the zone's energy generation (Supplementary Fig. 3). Our results show that hydropower availability significantly impacts the need for storage despite hydropower being responsible for less than 15% of the WECC's generation mix. Given that hydropower generation patterns are changing with the effects of climate change[33,34], researchers and

decision makers should use forward-looking climate and hydrological models to better capture hydropower availability when modelling LDES.

When analyzing different transmission expansion scenarios (scenario set C), we find that disincentivizing transmission expansion (by increasing the cost of new transmission lines ten-fold) only affects storage in a handful of load zones (Supplementary Fig. 4). Notably, Alberta's storage energy capacity increases by 474 GWh (+157%) and accounts for the vast majority of the WECC's 491 GWh increase in storage energy capacity (from 1.94 to 2.43 TWh). These results show that if transmission expansion is limited due to political, environmental, or other barriers, then the value of LDES could increase significantly in a handful of transmission-dependent load zones.

On the other hand, allowing unlimited transmission expansion by setting transmission expansion costs to zero significantly impacts the distribution of storage and generation resources in the WECC while having little impact on the WECC's total storage power and energy capacity. Setting transmission expansion costs to zero effectively

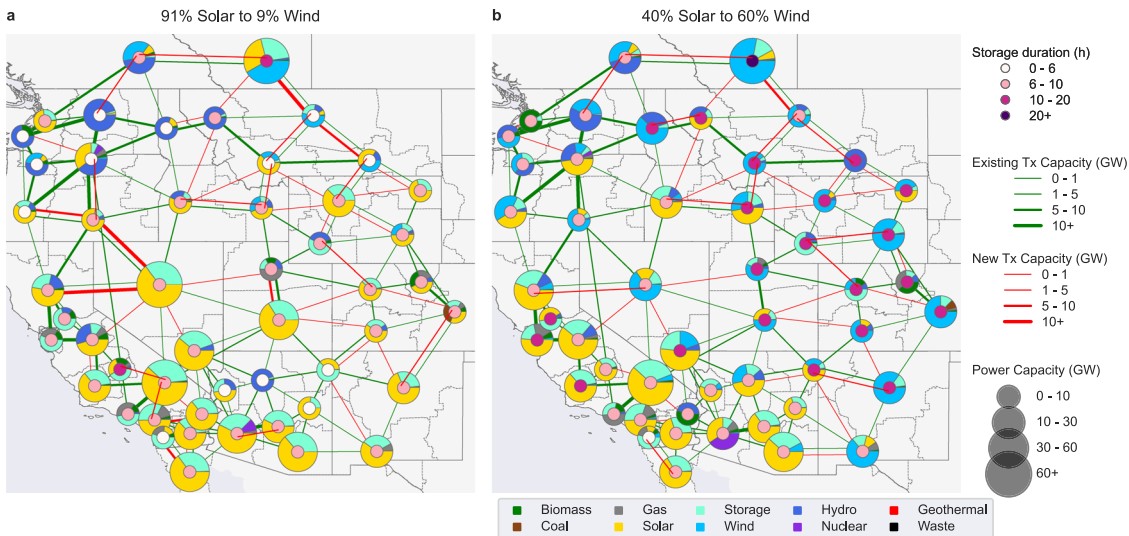

**Fig. 3 | Comparison of a solar-dominant grid vs. a wind-dominant grid.** Spatial comparison of the installed generation, transmission, and storage capacity across a zero-emissions Western Interconnect in a solar-dominated grid (**a**) compared to a wind-dominated grid (**b**). The pink inner circles represent the storage duration and highlight that solar-dominant regions tend to use 6-to-10-h duration storage while wind-dominant regions tend to use 10-to-20-h duration storage.

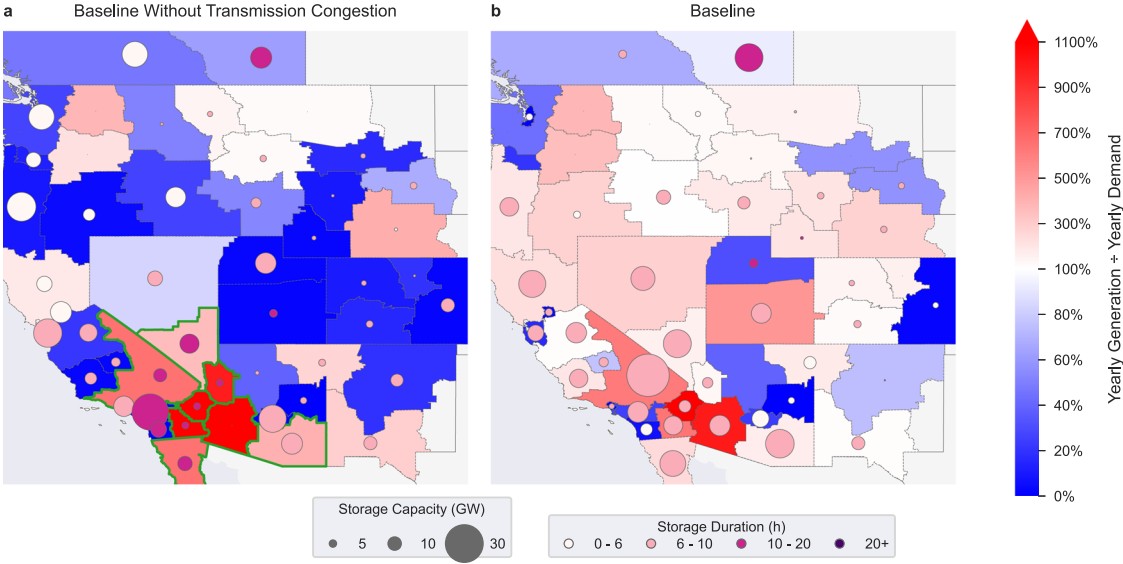

**Fig. 4 | Effect of eliminating transmission congestion on generation and storage.** Blue load zones generate less than their yearly demand (net importers), and red load zones generate more than their yearly demand (net exporters). Compared to the baseline (**b**), a Western Interconnect without transmission congestion (**a**) becomes reliant on eight load zones in the southwest (outlined in green) and most load zones (28 in 50) import power from neighbouring load zones to meet more than half of their yearly energy consumption. Circle size represents storage power capacity and circle color represents storage duration. The "without transmission congestion" scenario is equivalent to a scenario where there are no transmission expansion costs.

removes transmission congestion although transmission losses still occur. Without transmission congestion, generation shifts from the wind-dominant regions to the solar-dominated southwest. WECC-wide wind capacity drops by 54% (−53.9 GW) and solar capacity increases by 10% (+43.3 GW). Eight of the load zones located in the southwest (in green in Fig. 4) produce 70% of the WECC's total energy (compared to 35% in the baseline).

We find that varying the cost of storage energy capacity (scenario set D) is a significant driver of LDES deployment. This result matches those from Sepulveda et al.[25] who also identified that an energy capacity cost target of 1 US dollar per kilowatt hour ($/kWh) would fully displace firm low-carbon generation technologies. When varying energy storage costs from 102 to 0.5 $/kWh, the longest duration storage plants in the WECC vary from 8.9 h to 34 days. The 34 days (825 h) upper bound roughly matches the duration upper bound found in Dowling et al.'s simulations[24]. When energy storage costs are low, the increased LDES deployment is coupled with decreases in both wind capacity and new transmission installations (Fig. 2d). Table 4 summarizes these changes for different energy storage costs and Supplementary Table 1 provides California-specific values. Further, Supplementary Fig. 5 shows that when energy storage costs drop below 5 $/kWh storage is operated on seasonal cycles.

The significance and practical implications of these findings are considerable. Capacity expansion modeling can provide a technical optimum, however, capacity expansion in a real setting faces other considerations that can result in a grid that may deviate from what a

**Table 4 | Storage, wind, and transmission characteristics under varying storage energy capacity costs**

| Energy storage cost ($/kWh) | System-wide storage energy capacity (TWh) | System's median storage duration (h) | Largest storage duration (h) | Wind capacity (GW) | New transmission capacity (million MW-km) | System cost (billions of 2018 USD dollars) |
|---|---|---|---|---|---|---|
| 102 | 1.5 (−22%) | 6.3 | 8.9 | 113 (+14%) | 27 (+31%) | 130 (+24%) |
| 22 (Baseline) | 1.9 | 7.2 | 18 | 99 | 21 | 105 |
| 10 | 2.4 (+21%) | 8.5 | 29 | 98 (−1%) | 17 (−18%) | 100 (−5%) |
| 5 | 6.6 (+239%) | 19 | 378 (16 days) | 94 (−5%) | 13 (−40%) | 97 (−8%) |
| 1 | 22 (+1042%) | 127 (5.3 days) | 620 (26 days) | 82 (−17%) | 4.9 (−76%) | 86 (−18%) |
| 0.5 | 36 (+1747%) | 163 (6.8 days) | 825 (34 days) | 69 (−30%) | 5.3 (−75%) | 83 (−21%) |

Percentages in parentheses are the change compared to the baseline. For all scenarios, storage power capital costs are 19.58 $/kW and O&M costs of 6.10 $/kW-year. The baseline costs represent a scenario where the U.S. Department of Energy's "Energy Storage Grand Challenge" is achieved (90% reduction in storage costs by 2030).

modeling exercise can depict. Thus, the set of findings we discuss provide practical value as it reveals ideal LDES deployment that matches a myriad of possible futures that might be the result of political, social, environmental, and practical decisions affecting the grid. Transmission line expansion is highly sensitive and challenging in the USA. Hence, understanding where and how much LDES and imports would optimally support different zones as a function of how much transmission can be deployed are key considerations for planners. The findings around varying storage energy capacity costs are of particular interest for storage developers as they shed light on, for example, the duration that would become cost-effective as a function of the cost target their technology can achieve by 2050. This exercise is motivated by U.S. Department of Energy's "Energy Storage Grand Challenge" and "SunShot."

## The value and impacts of LDES mandates

In this section, we explore how the WECC would change if it had more LDES. An increase in energy storage could be achieved through policy, such as the implementation of LDES mandates[39]. Scenario set E compares the baseline containing 1.94 TWh of energy storage to 13 scenarios where the amount of energy storage is forced to be anywhere from 2 to 64 TWh. Figures 5 and 6 present the results of our analysis and show the impact of increased storage energy capacity on the grid and the marginal cost of electricity, respectively.

We find that solar and wind curtailment drops as up to 20 TWh if storage is mandated (Fig. 5a). The WECC's yearly renewable curtailment drops sharply from 118 GWh in the baseline to 9.6 GWh in the 20 TWh of storage scenario (−92%). Beyond this point, the impact is much more gradual. This sharp, then gradual decrease shows that the benefits of LDES in terms of reducing curtailment are most significant for the first 20 TWh of storage in the WECC. This 20 TWh mark plays an important role throughout the upcoming results.

We observe that storage decreases the need for transmission capacity and dispatchable renewables like biomass while shifting the solar and wind balance (Fig. 5b). Due to the significant drop in curtailment for scenarios up to 20 TWh, less generation capacity is needed to deliver the same energy to the grid. Hence, wind and solar capacity decrease which results in a 10% drop in the WECC's total generation capacity between the baseline and the 20 TWh scenario. Finally, transmission investments (measured in km-MW of lines installed) drop as storage energy capacity increases since transmission allows a region to meet its demand when generation resources are limited, and storage is an alternative way to meet that demand. Compared to the 1.94 TWh baseline, transmission investments drop by 30% in the 3 TWh scenario, 74% in the 20 TWh scenario, and 81% in the 64 TWh scenario. The ability of energy storage to reduce the need for transmission expansion is significant since transmission expansion is often challenging from a political and regulatory perspective.

We find that, beyond 4 TWh of storage mandates, storage is operated on bi-annual cycles and, beyond 20 TWh of storage mandates, storage is operated on yearly cycles. Beyond the 20 TWh

scenario, an additional yearly cycle (April to December) is superimposed over two seasonal cycles. As such, with more than 20 TWh of storage, the WECC-wide state of charge is near zero only during winter months (January through March).

We now focus our analysis on the impact of LDES on the marginal price of electricity. From our results we obtain 1.5 million marginal prices (one for each of the 14 scenarios, 50 load zones and 2184 time points) and analyze them by region, time of day and time of year.

We find that energy storage mandates largely reduce the variability in electricity prices, especially for the first 20 TWh of mandates (Fig. 6a). In the 1.94 TWh baseline, 82% of the marginal prices are at 0 $/MWh since for large portions of the year the WECC generates more renewable energy than it needs. The remaining marginal prices are high, with 11% of values above 200 $/MWh and 3% of values above 400 $/MWh. This large variability in marginal price decreases as energy storage is added to the grid since energy storage shifts the costs of generation during periods of peak demand to periods of low demand. For example, with 20 TWh of storage, 99% of marginal prices drop below 130 $/MWh and only 32% of marginal prices are still at 0 $/MWh. The median marginal price is 33 $/MWh.

We find that marginal electricity prices are lower in the southern WECC compared to the northern WECC and that energy storage mandates reduce marginal prices across all regions (Fig. 6b). Across all set E scenarios, the northern WECC (Canada, Oregon and Washington) has the highest marginal electricity prices, averaging 51 $/MWh in the baseline and 42–44 $/MWh beyond the 20 TWh scenario. The southern WECC (Baja Mexico, New Mexico, Arizona, Nevada) has the cheapest marginal electricity prices averaging 37 $/MWh in the baseline and 31 $/MWh in the 20 TWh scenario. The price gap between the north and southern WECC is likely due to the availability of cheap solar power in the south.

We find that marginal electricity prices are highest at night and that energy storage mandates reduce average marginal prices for all times of day (Fig. 6c). Across all set E scenarios, the average marginal price of electricity is 29% to 52% higher at night (8 p.m., midnight, and 4 a.m.) than at noon since cheap solar generation is not available during the night. We find that scenarios with more storage energy capacity have lower marginal electricity prices across all times of the day. Marginal prices drop on average 22% when moving from the 1.94 TWh of storage scenario to the 64 TWh scenario.

We find a significant difference in the marginal price of electricity for peak months compared to off-peak months. However, this price gap diminishes as energy storage is added to the grid (Fig. 6d). In the baseline scenario, July and December marginal electricity prices are highest at 180 $/MWh and 310 $/MWh, respectively, due to high demand during these months. As energy storage is added to the grid, the high July and December prices are reduced but prices in neighbouring months increase. In the 20 TWh scenario, average marginal prices for July, August, November, December and January range from 52 to 100 $/MWh while other months average 35 $/MWh or less. As more storage is added to the grid, variability in marginal prices across

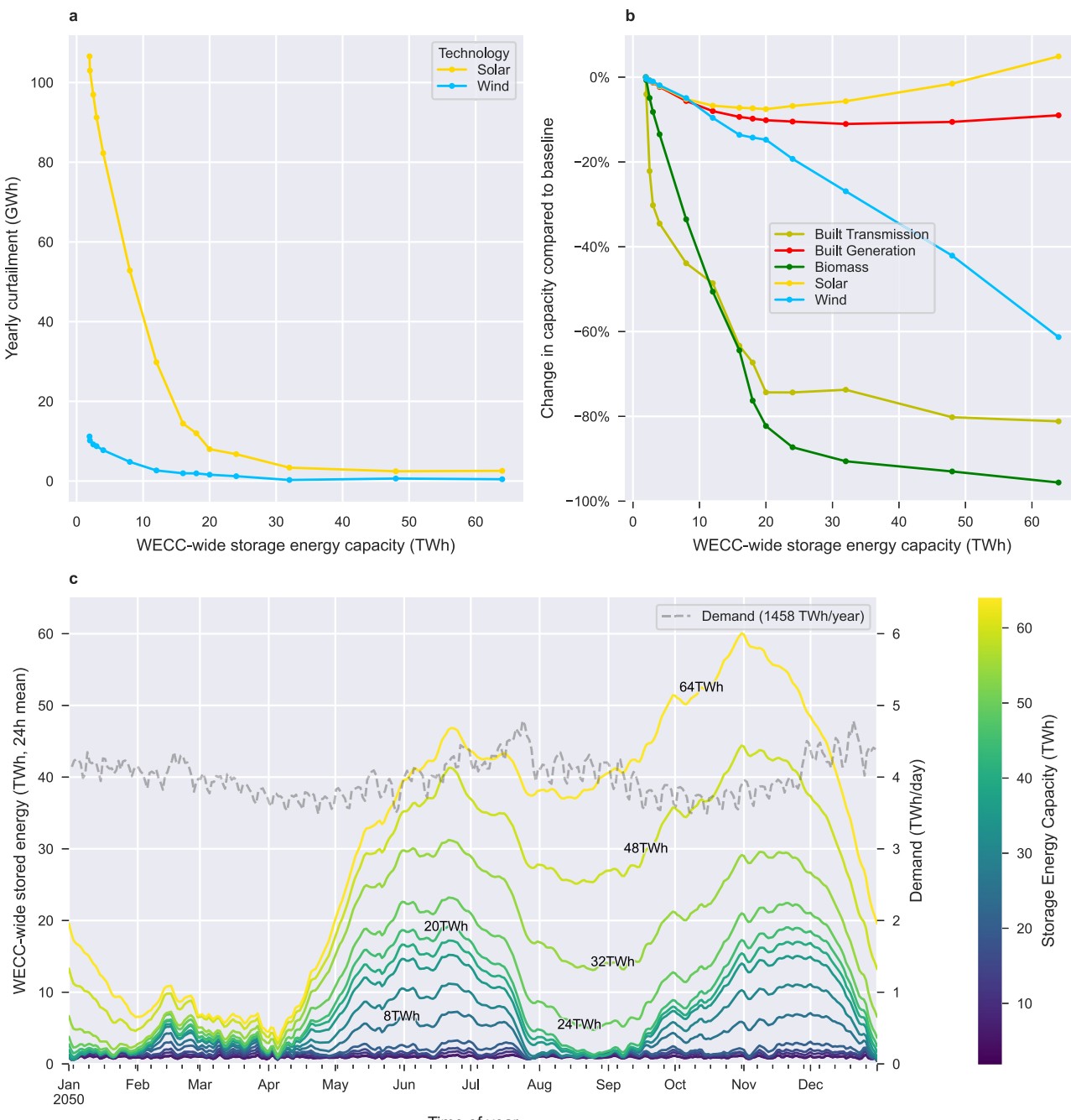

**Fig. 5 | Impact of long-duration energy storage mandates on curtailment, storage energy capacity and storage use.** Total changes within the Western Interconnect (WECC) in curtailment (**a**), generation capacity and transmission expansion (**b**), as well as energy held in storage (**c**) as the mandated amount (in TWh) of installed long-duration energy storage (LDES) increases. **c** Change in the quantity of energy held in storage across the WECC throughout the 2050 year. Operating storage on seasonal cycles first becomes cost-optimal at 4 TWh of mandated LDES, while operation on yearly cycle is first observed at 20 TWh of mandated LDES.

months further reduces. The higher costs in November, December and January correspond to the months of the year where storage is being discharged to near zero.

In summary, our results show that a 2050 decarbonized grid with greater storage energy capacity would reduce daily and seasonal variability in the marginal price of electricity while also reducing the marginal price of electricity across all regions and times of the day. As such, policies, subsidies, mandates or other events that would increase the penetration of storage resources in the WECC would likely result in lower prices in the wholesale electricity market while reducing price surges in July, December and night-time hours.

These results open a broad set of questions and considerations. First, some may argue that reductions in marginal prices and lower variability could trickle down as lower prices of electricity for consumers. However, these trends may also prompt the need for an electricity market and tariff redesign. There is a large body of work[44–50] that proposes solutions for this future decarbonized setting. Second, our results show that higher LDES mandates result in longer timeframes for energy arbitrage (even seasonal). In a real system, this would require the implementation of market signals (new ancillary services or a secondary long-term electricity market) for these assets to participate in and profit from seasonal arbitrage. Third, if an LDES

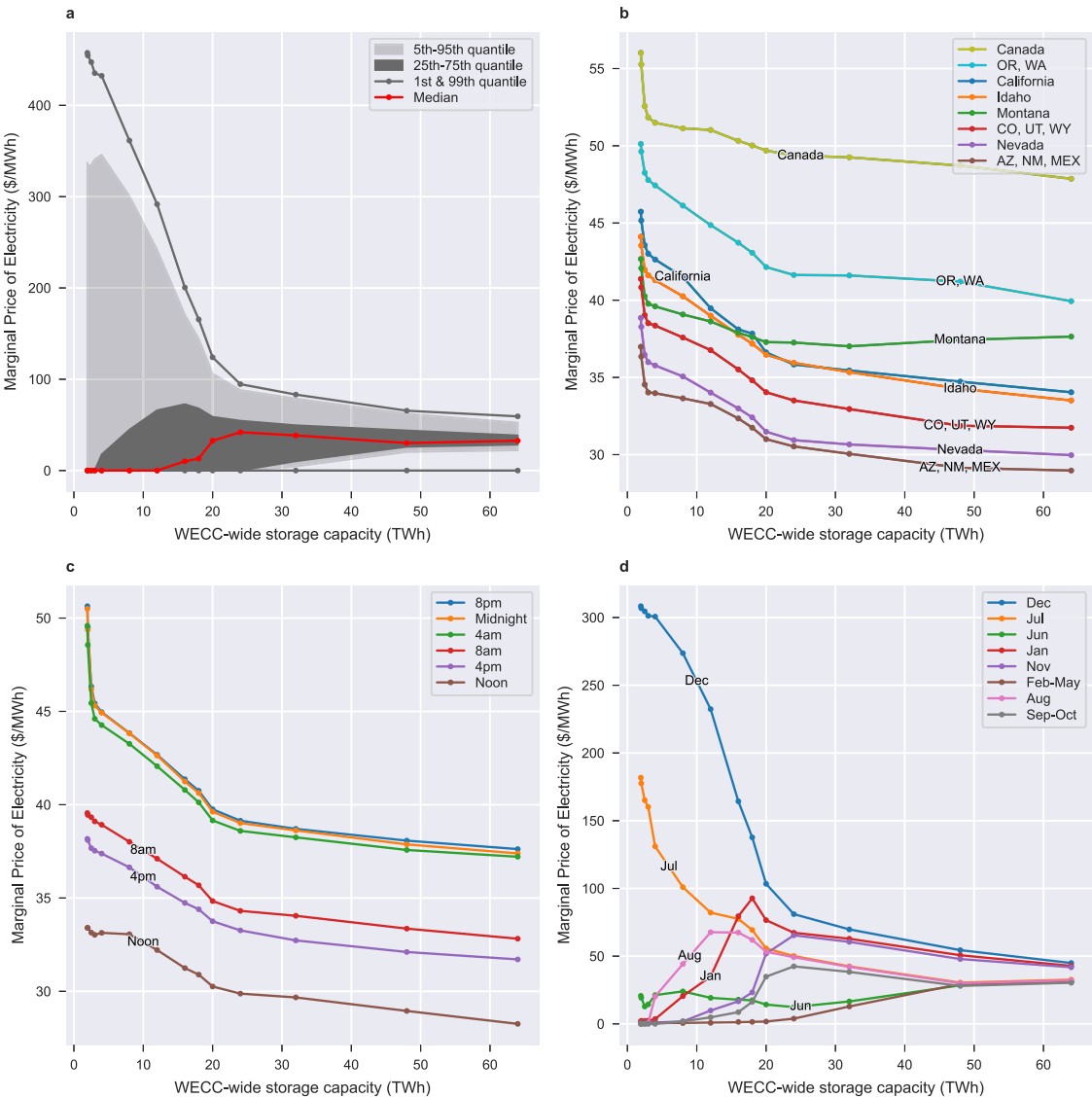

**Fig. 6 | Impact of long-duration energy storage mandates on the marginal price of electricity.** The total (**a**), regional (**b**), hourly (**c**), and monthly (**d**) distributions in the mean marginal electricity prices as the amount of mandated long-duration energy storage (in TWh) increases. Increases up to 20 TWh significantly decrease the variability in marginal prices while increases beyond 20 TWh have a lesser effect. Distributions stem from the marginal price of electricity in each load zone and at each timepoint (*n* = 109, 200).

mandate were to be passed at the state or interconnection level, this would provide a strong incentive for LDES developers to improve their technologies to participate in the new role our findings show these assets would play. Overall, in the past storage power capacity mandates have had an important impact; for example, the California Public Utilities Commission required the procurement of 1.3 GW of energy storage by 2020[51] and several states have followed this initiative[39]. These initiatives have had amplifying effects in solidifying the market for storage investors and suppliers. Futuristic LDES energy capacity mandates could have similar or stronger impacts for developers and would help to efficiently transition into a decarbonized grid as our results discuss.

Recapitulating, our analysis of 39 scenarios of a 2050 zero-emissions Western Interconnect explains the relationship between energy storage, electrical grid characteristics, and marginal electricity costs. First, our results suggest to industry and grid planners that the cost-effective duration for storage is closely tied to the grid's generation mix. Solar-dominant grids tend to need 6-to-8-h storage while wind-dominant grids have a greater need for 10-to-20-h storage.

Second, grid-modelling researchers should couple capacity expansion models with hydrological models that account for changing water flows under climate change since changes in hydropower availability would significantly increase the needed duration and capacity of storage. Third, our results suggest that policy makers should understand the relationship between transmission expansion and storage. If transmission expansion is hindered, the need for storage might increase significantly in a handful of transmission-dependent regions. Fourth, if energy storage capital costs drop below 5 $/kWh then extra-long duration energy storage (20–400 h) operated on seasonal cycles becomes cost-effective.

Further, increasing the storage energy capacity in the WECC through a mandate up to 20 TWh decreases the need for curtailment, and transmission expansion. Finally, increasing storage energy capacity in the WECC would reduce daily and seasonal variabilities in the marginal cost of electricity while also reducing the marginal cost of electricity across all regions and times of the day. These changes could translate into lower-cost electricity for consumers and lesser price surges.

## Methods

### Switch model formulation

The detailed Switch model mathematical formulation is given in the Supplementary Information. The model does not use relaxation variables (i.e., no constraint can be violated). All scenarios were solved using Gourbi's Barrier method with crossover disabled and default tolerances.

### Candidate generators

One of the key decision variables in Switch is the capacity investment of generation, out of a set of candidate generators with specific generating technologies and fuel sources, load zone locations, and other physical and financial generating characteristics. We use the dataset of candidate generators that was compiled in prior Switch-WECC analyses[30,41,42].

Candidate onshore and off-shore wind generators were derived based on wind power output from a gridded 3TIER Western Wind and Solar Integration Study dataset[8] and a gridded Canadian wind developer dataset, and a selection of prime sites based on criteria including high wind energy density, and proximity to transmission[30]. A portion of candidate generators were screened out in California if they were in "Category 3, high environmental risk" locations, which include areas legally excluded for development, protected areas with ecological or social value, conservation regions, and prime agricultural land[52].

Candidate solar generators include Residential PV (rooftop PV on homes), Commercial PV (rooftop PV on commercial buildings), Central PV (utility-scale), and Concentrating Solar Power with and without storage (solar thermal trough systems with or without thermal energy storage). Distributed Residential and Commercial PV candidate generation had been derived based on a gridded population density dataset, solar insolation data from NREL's (now deprecated) Solar Prospector tool, and assumptions on rooftop area and solar cell characteristics[30]. Available land and capacity for Central PV and Concentrating Solar Power candidate generators were screened based on land exclusion criteria (including national parks, wildlife areas, and steep terrain), solar insolation from the System Advisor Model from the National Renewable Energy Laboratory[53], and assumptions on the solar technology characteristics[30].

To simulate the dispatch of wind and solar generators, we use an exogenous dataset of hourly capacity factors by generator that had been constructed in prior Switch analyses[30,41]. For wind generators, hourly capacity factors for the candidate generator set were calculated from the 3TIER Western Wind and Solar Integration Study wind speed dataset[54] using idealized turbine power curves. For solar generators, hourly capacity factors for the candidate generator set are calculated from the System Advisor, using data from 2006 (consistent with the base weather year underlying the load profiles)[53]. Central PV and onshore wind generators with capacity-weighted average capacity factors below the 75th percentile for their technology were screened out to only have the candidate set among a computationally tractable, and commercially viable, set of higher-quality resource sites[30]. For existing solar and wind generators, we average the hourly capacity factors for all solar and wind generators, respectively, in each load zone, and assign all the generators in that load zone the average capacity factor for the given technology.

To obtain manageable computational complexities, we aggregate the utility-scale PV and onshore wind candidate projects throughout the WECC into one solar and one wind candidate project per load zone. The aggregation technique is as follows. In each load zone, we replace all the utility-scale PV candidate projects with one aggregated PV project whose parameters are identical to the candidate PV projects except for the following changes. (A) The total power capacity is the sum of the power capacities of each project. (B) The connection cost—representing the cost of new transmission needed to connect the plant to the grid—is the average connection cost of each project weighted by

the project's power capacity. (C) The variable capacity factors are found by analyzing the result of a previous baseline run completed without aggregation. The power generated (before curtailment) from utility-scale PV across the load zone is compared to installed PV capacity to obtain a single set of variable capacity factors for the entire load zone. Variable capacity factors for load zones without any installed utility-scale PV cannot be calculated and no aggregation is performed for these load zones. This sequence of steps is repeated identically for onshore wind candidate projects. We note that the aggregation provides only an approximate representation of the available solar and wind resources in a load zone. Future research could develop better solar and wind aggregation algorithms, better techniques for modelling at sparse hourly time intervals, or computational improvements that would enable greater model resolutions.

Biogas (from landfill, wastewater treatment plants, and manure) candidate generator availability is derived from an assessment of the technical resource/feed stock availability[55]. Bioliquid generators are allowed to be reinstalled in their current locations but no new bioliquid plants are assumed. No new biomass (bio solid) candidate generators are assumed, but cogeneration bio solid generation is allowed to be reinstalled at the end of its lifetime[30]. Candidate geothermal generators are based on the current locations and capacity of existing plants that may be reinstalled after retirement[30]. We assume there is no candidate hydropower generation.

Supplementary Table 2 summarizes the maximum capacity available for the set of candidate generators, and the capacity installed for existing generators.

We also note that our modelling of pumped hydro as well as residential and commercial PV is limited by Switch's implementation at the time of writing and the scope of the study. Specifically, the 24 pumped hydro plants in our model are modelled identically to 926 non-pumped hydro plants meaning that pumped hydro projects in Switch cannot draw energy from the grid as storage technologies do. The operation of each hydro plant is flexible and follows monthly averages of historical generation data and minimum generation requirements. Further, residential and commercial PV projects are treated as candidate projects in Switch which implies that system planners could decide how much residential and commercial PV to deploy. This may be possible through incentive programs, yet such programs and their costs have not been explored. An alternative approach would be to model residential and commercial PV growth as an exogenous assumption. Our future research will improve the modelling of pumped hydro and that of residential and commercial PV.

### Candidate storage

Table 2 summarizes the parameters for the candidate storage plants in the baseline model. In this paragraph, all mentions of NREL data refer specifically to NREL's 2020 annual technology baseline (ATB)[56] for utility-scale lithium-ion batteries. Although most parameters originate from NREL data for lithium-ion batteries, the parameters are meant to represent a technology-agnostic storage project. We choose to model technology-agnostic storage plants instead of multiple technology-specific storage plants to keep the focus of this study on the interactions between storage and the other components of the electricity grid. Baseline cost parameters represent a scenario where the U.S. Department of Energy (DOE) Energy Storage Grand Challenge is achieved. Specifically, costs are calculated by reducing NREL's 2020 costs for 10-h storage plants by 90% by 2030 per the DOE's target[57]. Costs are reduced such that the ratio of storage energy capacity costs to power capacity costs in a 10-h storage plant remains unchanged. Then, from 2030 to 2050, energy and power capacity costs are equally reduced by 25%—the 2030 to 2050 reduction rate projected by NREL's moderate case. Yearly O&M costs are set to 2.5% of the installation cost of a 10-h storage project. The value of 2.5% matches NREL's fixed O&M

cost projections[20]. We only model storage with 85% round trip efficiency and no idle losses. Future research could explore alternate storage technologies and their impacts on the grid.

## Demand modelling

Some papers in the literature model demand by simply scaling historical load curves using regional growth rates[23,26]. Other papers use limited demand scenarios that account for electrification but only to the extent that may be expected from the policies that already existed in law at the time of writing[17]. However, these approaches do not account for the ambitious decarbonization targets that will affect demand due to, for example, greater zero-emission vehicle adoption, building electrification, and energy efficiency advances[58]. In general, the lack of consideration for decarbonization targets was challenged in Jafari et al.'s review of energy storage literature[59].

In this paper, we use the "Compliant" demand scenario presented in Wei et al.'s study of 2050 energy scenarios for a low-carbon WECC[42]. This scenario complies with a decarbonization target of an 80% reduction in $CO_2$ emissions from 1990 levels by 2050. This target is the same as that of the demand model used in Sepulveda et al.'s "Electrification" scenario[25,60]. To achieve compliance, the Wei et al. scenario assumes a high rate of building energy efficiency retrofits, building electrification, and zero-emission vehicle adoption (see the Supplemental Information).

Wei et al.'s "Compliant" demand scenario is used as-is in our model and is not adjusted for variations in storage buildout across the scenarios presented in this paper. We acknowledge that demand and storage buildout are intrinsically linked (e.g., the model results show that less storage energy capacity leads to larger variations in electricity prices which might lead to less demand when prices are peaking). However, modelling the interplay between storage buildout and demand would require the simultaneous optimization of a capacity expansion model and a demand model. To the best of our knowledge, this has never been achieved while maintaining the geographical and temporal resolution presented in this paper. Even a recent 2023 study focusing specifically on the interplay between demand and energy storage used a two-step process where demand was first modelled and then used as an input to a capacity expansion model[61].

Additionally, due to the limited scope of our paper, we do not model demand-side management (i.e., demand response) or potential cross-sectoral grid components[62] like the production and consumption of hydrogen and management of electrical vehicle charging. Our team's future research will explore these topics to gain a broader understanding of the role of LDES in the WECC.

## Discussing the effects of a 4-h temporal resolution

Our model temporal resolution is limited due to the computational complexity of modelling the WECC (5918 plants post-aggregation) over 365 days. Despite using a powerful server (32 cores, 2.5 GHz, AMD EPYC 7502 P, 512 GB RAM), it was only feasible to run all 39 scenarios with a 4-h temporal resolution (the energy balance was calculated at 12 am, 4 am, 8 am, 12 pm, 4 pm, and 8 pm on each day). Using a 4-h temporal resolution differs from the 1-h temporal resolution that is commonly used in literature. However, studies that use a 1-h temporal resolution for 365 days are forced to make other computational trade-offs such as not modelling transmission[24,25], not modelling existing generation capacity[24,25], or optimizing storage buildout and storage dispatch separately[26]. Given the focus of the study on the interaction between LDES and the WECC grid, we judged that such trade-offs would be worse than reducing the temporal resolution to 4-h.

To understand the impact of the 4-h temporal resolution on our results, we compare our baseline (4-h resolution) to a supplementary scenario that uses a 1-h temporal resolution (and no wind or solar plant aggregation). We note that although it was feasible to solve (within 13 h) the baseline scenario at a 1-h resolution, experience indicates that

not all 39 scenarios in this study could have been solved to optimality at this resolution.

Upon comparing 13 key results relating to curtailment, generation capacity, demand, storage buildout, and transmission (Supplementary Table 3), we find that the 4-h baseline requires less resources than the 1-h scenario. The 1-h scenario shows an increase in the power capacity of storage (+18.2 GW; +7.6%), solar (+11.6 GW; +2.7%), wind (+10.1 GW; +10.2%), biomass (+1.7 GW; +18.7%), and transmission (+8.2 million MW-km, +6.6%). Across all technologies, storage and generation capacity increases by 4.9%. As such, the numerical results presented in this paper have a bias towards underestimating the required capacity. However, this bias is less than 10% for nearly all technologies. A bias of this magnitude is acceptable since it is not any greater than the uncertainties inherent with predicting a 2050 future (e.g., uncertainties in technology costs). Despite this bias, we expect that the discussed trends and insights hold true given that the storage duration, storage energy capacity and demand for both scenarios vary by less than 3%. Further, the average daily dispatch curve (Supplementary Fig. 6) has a similar shape between both the 4-h and the 1-h scenario which indicates a good selection of hours in the 4-h scenario (e.g., the inclusion of 12 pm in the 4-h scenario ensures that time of peak generation is represented in the model).

## Limitations and future research

In addition to the previously mentioned areas of future research, we highlight four areas where our modelling is limited and space for future research exists.

First, the model does not include connections to other interconnects (e.g., ERCOT). Some studies have noted the benefits of greater coordination between interconnects as such coordination provides flexibility during dispatch which displaces the need for alternative more expensive sources of flexibility such as LDES[17,18]. However, interconnect coordination faces significant sociopolitical hurdles common to large transmission expansion projects[63]. To avoid basing the entirety of this paper on a potentially socio-politically infeasible future, and given the scope of the study, we did not include connections to other interconnects in the model. We aim to understand the potential of interconnect coordination on the value of LDES in a future study. Further, we encourage policy makers to engage with socio-politically challenging solutions such as interconnect coordination since ambitious action is needed to meet climate change targets[1].

Second, our results are limited by our ability to predict future weather trends and energy demand. As discussed in this paper and other research[64,65], the value of LDES is closely tied to the seasonal patterns in energy demand and renewable generation. A 2050 future with significantly different demand data or variable capacity factors data may lead to different results. To gain a fuller picture of the role of LDES, future research should further examine the modelling assumptions behind our demand and weather data while exploring alternate scenarios.

Third, due to combined limitations in data availability, the scope of the study and computational complexity, we do not model energy reserve requirements, nor do we run simulations for "extreme" years where solar or wind power is more intermittent than usual. Doing so in future research would be key considering that LDES energy storage would likely be more favourable when considering energy reserve requirements or when renewable generation is limited.

Finally, numerous papers have discussed the benefits of firm generation such as carbon capture and sequestration (CCS) technologies[13–16,21,66,67]. In this paper, we include some firm generation technologies in the model (i.e., biomass and geothermal) but do not include CCS technologies. The reasoning behind this decision is twofold. First, Sepulveda et al. have already studied the impact of CCS and other upcoming firm generation technologies specifically with respect to LDES[25]. Second, although CCS technologies are expected

to play a role in a 2050 future, it is still unclear what this role will be. Currently, government pledges regarding carbon removal are far below what is needed to meet the Paris agreement[68] leading some to emphasize that CCS should only be a solution for hard to abate sectors like aviation, not an alternative to decarbonizing the electrical grid[69].

## Reporting summary

Further information on research design is available in the Nature Portfolio Reporting Summary linked to this article.

## Data availability

An archive containing all the data used to generate this paper's figures as well as all the data used and produced in the baseline scenario (i.e., the CSV input and output files) is hosted on GitHub at https://github.com/REAM-lab/Staadecker_et_al_2024_archive. This archive also contains the configuration files for the other scenarios in the study as well as a README.md that provides instructions detailing the contents and structure of the archive.

## Code availability

The code that defines our model, handles all data processing, and generated the figures in this paper is available at https://github.com/REAM-lab/switch/releases/tag/v2.0.0.

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

## Acknowledgements

The authors would like to thank Matthias Fripp, Josiah Johnston, Rodrigo Henriquez-Auba, Benjamin Maluenda, Ana Mileva and Jimmy Nelson for their prior contributions and developments of the Switch model. Special thanks go to Matthias Fripp and Rodrigo Henriquez-Auba for their courtesy of sharing the mathematical formulation Latex files of their supplemental information for us to continue expanding upon. M.S. thanks the Division of Engineering Science at the University of Toronto for financial support through the Engineering Summer Research Opportunity Program. Additionally, this work was partly supported by the California Energy Commission [EPC-19-060]. This document was prepared as a result of work sponsored by the California Energy Commission. It does not necessarily represent the views of the Energy Commission, its employees, or the State of California. The Energy Commission, the State of California, its employees, contractors, and subcontractors make no warranty, express or implied, and assume no legal liability for the information in this document; nor does any party represent that the use of this information will not infringe upon privately owned rights. This report has not been approved or disapproved by the Energy Commission nor has the Energy Commission passed upon the accuracy of the information in this report.

## Author contributions

Conceptualization, M.S. and P.H-G.; Software, M.S., P.A.S-P., J.S., and P.H-G.; Investigation, M.S.; Data Curation J.S. and P.H-G.; Writing – Original Draft, M.S. and P.H-G.; Writing – Review & Editing, M.S., S.K., J.S., and P.H-G.; Visualization, M.S.; Supervision, P.H-G.; Funding Acquisition, P.H-G. and S.K.

## Competing interests

The authors declare no competing interests.
