## [Peer Review File · Nature Communications]

REVIEWER COMMENTS

Reviewer #1 (Remarks to the Author):

I enjoyed reviewing this article. It adds a new angle to the LDES debate that I have not seen elsewhere. I support its publications

Reviewer #2 (Remarks to the Author):

COMMENTS / QUESTIONS FOR THE AUTHORS:

Very important study and well written article. Here are some comments for consideration that speak to clarity in the findings of the study, rather than quality of the article itself.

1). In the abstract of the article it is stated that the model discussed has been applied "to zero-emissions Western Interconnect with high geographical resolution." I think it would be good to include a statement in the abstract about the applicability, or lack thereof, of the research findings to regions outside the Western Interconnect.

2)The authors establish how LDES is defined for the purpose the study, but references to short-duration storage do not appear to be defined. While the inference may be there by comparing against the definition of LDES, it may be useful to clearly state a definition for short-duration storage.

3) The term "weather year" is unfamiliar to me and I would recommend that it be defined.

4) The introduction to the article makes a reference to "LDES technologies" but it is not until the conclusion that it is established that the study has taken a technology-agnostic approach. I think it would be good to establish this earlier in the article.

5) I believe the references to mandates need to be further defined. While 11 states have a storage procurement mandate, target or goal, as far as I am aware there is no state that has a procurement mandate for LDES. California has done modeling to determine the amount of LDES that will be needed, but as far as I know this has not been reflected into a procurement mandate specific to LDES. Thus, it's unclear to me and likely to other readers how procurement policies that are specific to short-duration storage have a direct correlation to the study of LDES.

6) The study examines "the impact of decreases in hydropower generation on LDES." If this study is specific to the Western Interconnect region, to what extent is there applicability to other regions that do not have historic levels of hydropower that can be studied to show the impact of decreased hydropower in those regions?

7) Related to "Figure 5. Impact of LDES Mandates on Curtailment, Storage Energy Capacity and Storage Use," the authors make the statement that they found that "solar and wind curtailment drops as up to 20 TWh of storage is mandated." Again, I would offer that known storage mandates are associated with short-duration energy storage, so the relevance is perhaps questionable and the implication that LDES mandates exist and are a quantifiable factor may be a bit misleading.

Reviewer #3 (Remarks to the Author):

This article presents a very interesting look into an uncertain grid future bounded by our existing infrastructure and geographical energy mixes. The article is well written and considers the overall complexity of changing climate, economics, grid makeup, and generation makeup.

Similar to many studies in the Long Duration Energy Storage field, this effort will serve as an initial step to be refined over time as generation mixes and load evolve. This article is clear about the limitations, specifically in running scenarios with four hour intervals.

Overall the methodology appears sound. Specific comments for clarification are included in the marked up manuscript. Nice work by the team that developed this analysis!

Reviewers response for NCOMMS-23-48089 “The Value of Long-Duration Energy Storage under Various Grid Conditions in a Zero-Emissions Future”

Martin Staadecker, Julia Szinai, Pedro A. Sánchez-Pérez, Sarah Kurtz, and Patricia Hidalgo-Gonzalez

We would like to thank the reviewers for their valuable feedback in this review round. In what follows, we provide detailed responses to their comments.

Reviewer #1 (Remarks to the Author):

I enjoyed reviewing this article. It adds a new angle to the LDES debate that I have not seen elsewhere. I support its publications

Answer: Thank you so much for your review and time spent. We are very glad to hear you found the work insightful.

Reviewer #2 (Remarks to the Author):

COMMENTS / QUESTIONS FOR THE AUTHORS:

Very important study and well written article. Here are some comments for consideration that speak to clarity in the findings of the study, rather than quality of the article itself.

Answer: Thank you so much for your review and support of our work. We deeply appreciate the comments provided to clarify some of our findings.

1). In the abstract of the article it is stated that the model discussed has been applied "to zero-emissions Western Interconnect with high geographical resolution." I think it would be good to include a statement in the abstract about the applicability, or lack thereof, of the research findings to regions outside the Western Interconnect.

Answer: Thank you for your suggestion. We changed the following sentence from the abstract: “Given the asset and resource diversity of the Western Interconnect, our results can provide grid planners with guidance on how LDES impacts and is impacted by energy storage mandates, investments in LDES research and development, and generation mix and transmission expansion decisions.”

2)The authors establish how LDES is defined for the purpose the study, but references to short-duration storage do not appear to be defined. While the inference may be there by comparing

against the definition of LDES, it may be useful to clearly state a definition for short-duration storage.

Answer: Thank you for your comment. We have now included a definition for short-duration storage: “In this paper, we follow the emerging trend^{31,32} of defining LDES as any type of storage with 10 or more hours of duration. Conversely, short-duration storage is defined as any type of storage with fewer than 10 hours of duration. We also define seasonal storage—a subset of LDES—as any type of storage that is operated such that charge-discharge cycles occur over several months.”

3) The term "weather year" is unfamiliar to me and I would recommend that it be defined.

Answer: Thank you for noticing we did not define it. Now we have clarified what is a weather year the first time we use the term: “They find that LDES duration increases from ~400h to ~700h as more years of weather data, i.e., weather years, are considered.”

4) The introduction to the article makes a reference to "LDES technologies" but it is not until the conclusion that it is established that the study has taken a technology-agnostic approach. I think it would be good to establish this earlier in the article.

Answer: Thank you for noticing this. We have clarified this when we describe the research gaps we are addressing: “To address these research gaps, this study conducts a systematic analysis that identifies and studies four of the most important characteristics of the grid that affect the value and optimal deployment and operations of technology-agnostic LDES in a zero-emissions power system.”

5) I believe the references to mandates need to be further defined. While 11 states have a storage procurement mandate, target or goal, as far as I am aware there is no state that has a procurement mandate for LDES. California has done modeling to determine the amount of LDES that will be needed, but as far as I know this has not been reflected into a procurement mandate specific to LDES. Thus, it's unclear to me and likely to other readers how procurement policies that are specific to short-duration storage have a direct correlation to the study of LDES.

Answer: Thank you for bringing this up. We realized we need to better motivate and explain what we are intending with that set of scenarios. The study of “prospective” or “potential” LDES mandates does not mean to imply they are already in place or even being considered in some region. We believe in the importance and benefits of evaluating the design of possible future LDES energy capacity mandates as we further decarbonize our grids. This is why we decided to include this set of scenarios where we tested the impacts and benefits of different theoretical LDES energy capacity mandates. We want to bring to policy makers’ attention the infrastructure, operational, and electricity pricing benefits of considering LDES energy capacity mandates. We contextualized this set of scenarios as being motivated by current short-duration storage procurement mandates on power capacity. We thought there could be interesting insights and results from testing an

alternative, but related, policy instrument as an energy capacity mandate. We hope our work can inspire further studies from system operators and public utility commissions in this direction as it can become a valuable policy instrument moving forward –similarly to how storage power capacity mandates have been valuable in our early stages of decarbonization.

We believe we have now clarified this in a few places through the manuscript:

“Fourth, there are no prior studies that analyze the impact and potential benefits of LDES energy capacity mandates. In this context, we refer to an LDES energy capacity mandate as a quantity of storage energy capacity that is mandated by a governmental entity to be built by 2050 across Western North America. Power capacity storage mandates have had an important role; for example, California was the first state to have power capacity storage mandates to support grid decarbonization^{38,39}. This initiative has had multiplicative effects in solidifying the market for storage investors and suppliers, encouraging more R&D, and producing operational benefits for the grid as more renewables are integrated. Several states have since followed this initiative⁴⁰. In this study, we focus on evaluating the design of possible future storage energy capacity mandates instead of power capacity mandates because we want to understand the energy balancing benefits of such mandates and their impacts for the grid (e.g., electricity pricing impacts, curtailment and operational impacts, zonal distribution of optimal LDES placement, etc.). To the best of our knowledge, this is a research question relevant for policy makers that has not yet been explored.”

“Overall, in the past storage power capacity mandates have had an important impact; for example, the California Public Utilities Commission required the procurement of 1.3 GW of energy storage by 2020³⁹ and several states have followed this initiative⁴⁰. These initiatives have had amplifying effects in solidifying the market for storage investors and suppliers. Futuristic LDES energy capacity mandates could have similar or stronger impacts for developers and would help to efficiently transition into a decarbonized grid as our results discuss.”

6) The study examines "the impact of decreases in hydropower generation on LDES." If this study is specific to the Western Interconnect region, to what extent is there applicability to other regions that do not have historic levels of hydropower that can be studied to show the impact of decreased hydropower in those regions?

Answer: Thank you for asking this important question. In regions outside the Western Interconnect without hydropower generation, the optimal LDES baseline deployment for a decarbonized grid may align with the results we observed for a number of solar-dominant or wind-dominant Western Interconnect sub-regions (as these are the predominant sources in most future zero emissions grids). There are several regions (load zones) in our model that do not have hydropower generation, and arguably, may not even be supported through hydropower imports. This set of regions could be looked into in more detail to see how LDES gets deployed depending on the rest of the other factors (i.e., transmission expansion constraints, solar versus wind dominance, etc.).

Alternatively, our scenarios with decreases in hydropower generation can also be considered more generally representative of a future when additional LDES must be built to compensate for a loss of flexible, zero-emissions resources. For example, in regions around the world that have more geothermal or biomass resources instead of hydropower, a loss of those flexible, zero-emissions resources would be analogous to a decrease in hydropower because it would need to be replaced by additional LDES.

We have added a clarification to the description of Set B in Table 2:

“This set with decreases in hydropower generation can also be considered more generally representative of a future when additional LDES may need to be built to compensate for a loss of flexible, zero-emissions resources.”

7) Related to "Figure 5. Impact of LDES Mandates on Curtailment, Storage Energy Capacity and Storage Use," the authors make the statement that they found that "solar and wind curtailment drops as up to 20 TWh of storage is mandated." Again, I would offer that known storage mandates are associated with short-duration energy storage, so the relevance is perhaps questionable and the implication that LDES mandates exist and are a quantifiable factor may be a bit misleading.

Answer: Thank you for bringing this up in the context of this figure. We agree we do not intend to be misleading by implying the current existence of LDES mandates. Thanks to your feedback we have now clarified that this is an exploration of potential novel and future policy tools that could help reach higher levels of decarbonization (similarly to the role storage power capacity mandates have aided the early stages of decarbonization).

Reviewer #3 (Remarks to the Author):

This article presents a very interesting look into an uncertain grid future bounded by our existing infrastructure and geographical energy mixes. The article is well written and considers the overall complexity of changing climate, economics, grid makeup, and generation makeup.

Similar to many studies in the Long Duration Energy Storage field, this effort will serve as an initial step to be refined over time as generation mixes and load evolve. This article is clear about the limitations, specifically in running scenarios with four hour intervals.

Overall the methodology appears sound. Specific comments for clarification are included in the marked up manuscript. Nice work by the team that developed this analysis!

Answer: Thank you so much for your detailed review and supportive comments. We have implemented your suggestions in our manuscript.